# Application of Hierarchical Colored Petri Nets for Technological Facilities' Maintenance Process Evaluation

**Sergey P. Orlov *** , **Sergey V. Susarev and Roman A. Uchaikin**

Institute of Automation and Information Technology, Samara State Technical University, 443100 Samara, Russia; susarev_sergey@mail.ru (S.V.S.); uchaykinra@yandex.ru (R.A.U.)
* Correspondence: orlovsp1946@gmail.com

**Abstract:** The high reliability of modern engineering systems is achieved by performing predictive maintenance. Mathematical models based on stochastic timed colored Petri nets are an effective tool for developing complex production processes for Industry 4.0. This article discusses the maintainability evaluation used in hierarchical Petri net models. The hierarchical simulation model was built using timed colored Petri nets, and was constructed with four levels of repair and maintenance modules. New module structures are proposed for simulating the schedule of production tasks and interaction with technological units. The emphasis is on the processes of predicting maintenance and repair, moving units to service, replacing units, and forming a reserve. The design of the simulation modules allows the setting of probabilistic parameters for the distributions of equipment failures, requests for unit maintenance, repair time, and recovery time after repair. The article proposes to use the hierarchical Petri model in conjunction with solving the problem of minimizing the cost of service. The iterative procedure consists of obtaining an approximate unit distribution by tasks, subsequent simulation of the technological process, and adjusting the optimization problem constraints. For example, the hierarchical Petri net is considered to assess the maintainability of autonomous agricultural vehicles. The results of the simulation experiments are presented. A simulation of the agrotechnical production process was performed, during which vehicles were maneuvered, taken out for repair or maintenance, and returned to the reserve fund. The interdependencies of preventive maintenance periods, service operations, failure rates, and predictive maintenance requests were obtained in order to comply with the task scheduling. The proposed model is a generalization, but it is especially effective in studying mobile equipment servicing.

**Keywords:** model-based development; timed colored Petri nets; predictive maintenance; maintainability evaluation; discrete-event simulation

## 1. Introduction

The development of technologies in industry moves along the path of complicating technological operations, creating flexible and robotic industries. At the same time, the requirements for the reliability of technological equipment are increasing. There is a trend towards a shift from preventive maintenance to predictive maintenance of equipment, which corresponds to the general direction of technological process development according to the concept of Industry 4.0 [1,2].

Today, virtual testing and virtual commissioning based on the behavior simulation of production systems are widely used. Such technologies rely on a model-based development [3,4]. At this stage, the studies of maintenance and repair processes become essential, as they provide the organization of the predictive maintenance system at the time of actual product output. Another important aspect is the continuous monitoring of technological equipment in the production process to replace parts or serviced units in a timely manner, predicting failure or defect.

For many applications in industry, discrete-event simulation is the most appropriate tool for studying technological processes and equipment properties. Many kinds

of parts create specific difficulties in describing the models of machines, components, and assemblies. Due to the stochastic nature of the technological process, calculating unit performance is impossible. These circumstances lead to Petri nets' (PNs') mathematical formalism, aiming to build simulation models of production processes and maintainability evaluation [5,6].

This article discusses a method for constructing a hierarchy of simulation models based on timed colored PNs, describing processes in a set comprising:

- Performed production tasks, specified in the form of a Gantt chart;
- A set of technological units;
- Service stations for maintenance and repair.

This article contains the following parts: Related works are presented in Section 2. The problem statement of evaluating maintenance and repair efficiency is discussed in Section 3. Section 4 briefly describes the definitions of colored Petri nets (CPNs) and timed colored Petri Nets (TCPNs). Section 5 presents the hierarchical Petri net model (HPN), and describes how this architecture is abstracted and modeled using TCPNs. Sections 6 and 7 provide an example of the use of HPN to simulate an autonomous agricultural vehicle system and discuss the results obtained. The future extensions to the proposed approach are presented in Section 8.

## 2. Related Works

Models based on Petri nets have been used to describe discrete-event systems with the parallel–serial nature of the processes. The reason for choosing PNs is that, unlike traditional models for modeling discrete events (DEs), the PN model provides an intuitive graphical representation of the studied complex system, and has been proven to be more flexible and efficient [7–9]. Among the many practical applications of Petri nets, works related to the modeling of flexible manufacturing systems (FMSs) stand out. Early work has shown the effectiveness of various Petri nets for simulating production systems for defect and failure prediction [10–12]. In the subsequent period, new modeling tools for studies of production systems were built, which describe the process of equipment degradation and deterioration [13,14]. In [12,15], high-level Petri nets were developed, corresponding to the hierarchical organization of production structures. This explains the continuing interest of researchers, which has led to the creation of various extensions of Petri nets. As a result, the most widely used models are colored CPNs [16,17], timed TCPNs [18,19], and stochastic SCPNs [20].

Another relevant area of Petri net model application is in developing, testing, and verifying software systems, computational processes, and communication protocols. Mainly, the successful application of Petri nets should be noted in the description of information processing, microprocessor systems, and others.

In such systems, there are strict requirements for fault tolerance and timing relationships. Kumar et al. [21,22] applied CPNs to formalize component operation scheduling, and presented a method for analyzing system dependencies based on a service-oriented architecture (SOA). The various types of dependency relationship are formalized using service algebra. Such a model-driven approach provides a process for reworking and restructuring the software application. Del Foyo and Silva [23] developed a similar approach for requirements analysis in real-time systems. They proposed a reachability-based algorithm for real-time testing of discrete systems, in which timed computational tree logic is used as a specification language, and timed transition graphs are introduced to represent system behavior.

Among a wide range of topics for the practical use of PNs, one should highlight the construction of stochastic Petri nets for studying the degradation, aging, and deterioration processes of various units, mechanisms, and machines. The study of these phenomena and processes is necessary for the organization of maintenance and repair. A review of the existing research in this area shows that many researchers successfully use the

apparatus of Petri nets to design reliable and maintainable industrial systems in various areas of technology.

Preventive and predictive maintenance strategies are used in technological processes; however, discrepancies exist between the mathematical models used and the actual maintenance activities. One of the main reasons for this is insufficient mathematical modeling methods to accurately describe the equipment interactions during maintenance. In this regard, the use of simulation models on Petri nets is actively developing.

The relevance of modeling the interactions between units and component manufacturing processes causes the active development of Petri nets in this area. Boukredera et al. [24] developed color PNs as a formalization of the temporal aspects of the elements functioning in a decentralized task distribution. Simultaneously, the presentation of the object as a multi-agent system is combined with the CPN model simulation. The CPN model for production–logistics problems, which implements the self-adaptive collaboration method, was investigated by Guo et al. in [18]. The objective of the proposed TCPN model is to minimize the waiting time of all jobs, the makespan, and the electricity consumption of all machines and AGVs. Three key performance indicators, including waiting time, makespan, and electricity consumption, are considered. Pla et al. suggested a new Petri net extension for modeling workflow activities together with their required resources: resource-aware Petri nets (RAPNs) [25]. Workflows are modeled with Petri nets, since they are a well-known tool for workflow modeling, and offer a wide range of extensions to facilitate this task. In the article, five different scenarios were considered for testing the latency of the maintenance prediction procedure. Each scenario includes a specified number of jobs and different resources depending on the source of the workflows.

Technological processes are stochastic due to the impact of some external random factors. Accordingly, many kinds of research have been conducted on creating Petri nets to simulate random events with a given probability density distribution. Various modifications of stochastic colored Petri nets (SCPNs) have been developed for these purposes. In [26], Santos et al. developed generalized SCPNs with predicates for Monte Carlo simulation. The times to component failure are considered to be the Weibull distribution. Marsan [27] proposed to simulate random processes at PN transitions using delays with negative exponential probability density functions. Stochastic colored Petri nets are used for a comprehensive class of objects, particularly when assessing maintainability and reliability. Lu et al., in [17,28,29], also investigated the application of SCPN tuples and the calculation of the required resources according to the equations of the model state. Consumable and reusable resources are represented as multisets in SCPN places. The simulation of maintenance allows for the estimation of resource requirements at each stage.

There are known examples of the effective use of SCPNs in studying the maintenance of an electric car-sharing system [30], researching the degradation and deterioration of ceramic claddings [31], and modeling cyber–physical systems [32]. The simulation models are developed, integrating physics-based and dependability models in order to monitor the state evolution of dam protection from natural phenomena and improve maintenance decision-making processes [33].

Researchers at the University of Nottingham, UK, actively work on the theory and practical use of simulation models for railway infrastructure. Le, Andrews, and Fecarotti in [34–36] presented a bridge model developed based on the Petri net (PN) approach. Their method allows for detailed modeling of the individual components in the structure, while maintaining the analytical problem at a manageable size, resulting in an efficient analysis. The bridge model is formed from sub-models of each of the bridge components, and considers the component deterioration process, the interactions and dependencies between different component deterioration processes, and the inspection and maintenance processes. The model states are defined based on the actual degraded component conditions that they experience.

In [37], several types of Petri nets with different properties were introduced. The extension of Petri nets, adaptive Petri nets (APNs), which satisfy the specific requirements of

railway rolling stock maintenance, was proposed. The authors developed an application of APNs in combination with Monte Carlo simulations to evaluate the maintenance of railway rolling stock.

The development of these approaches has continued with the creation of offshore wind turbine models. Le and Andrews in [38] used stochastic Petri nets. This method's versatility made it possible to include details of degradation and maintenance operations in the model. Dependent deterioration processes between subsystems of wind turbines were considered, defining the maintenance rules for early indication of failure to ensure replacement. Leigh, Yan, and Dunnet produced simulation models of offshore wind turbines [39,40]. Their model is based on SCPNs, describing three types of maintenance: periodic, conditional, and corrective maintenance. The wind turbine conditions were monitored to determine the frequency of service, and then the repair and maintenance processes were simulated.

On the other hand, one more positive property of CPNs should be noted: implementing the hierarchical models of complex systems. The functionality of many modeling systems that support Petri nets allows the realization of both top-down and bottom-up designs. Lorbeer [41] proposed reconfigurable Petri nets based on a hierarchical principle. Sheng et al. [42,43] developed hierarchical CPNs to simulate the air fleet service. These papers presented hierarchical models of a fleet operation and maintenance process, which considers mission-oriented operation, multiple-level maintenance, multiple cannibalization policies, maintenance schedule, and spare inventory management.

One work by Baruwa and Piera [44] was devoted to managing autonomous vehicle schedules, which is closest to the area of our research on autonomous agricultural vehicle maintenance. The attention of researchers has also been paid to the planning of mobile robot movement [45], human–robot interaction [46], and the positioning of autonomous vehicles on territory [47]. However, this work did not consider the procedures for servicing autonomous cars based on failure and defect prediction.

## 3. Organization of the Maintenance and Repair of Technological Process Units

We assume that the main component of the technological process is the "Task." The unit of equipment used in the technological process is denoted as "Unit." We consider a set of technological tasks $z_k \in Z, k = \overline{1,K}$, a set $AU$ of available units $u_n \in AU, n = \overline{1,N}$, and a Gantt chart $G_t$, which describes the schedule for performing technological tasks.

The maintenance and repair management system should perform the following functions:

1.　The model assigns available units $u_n$ for tasks according to the schedule $G_t$.
2.　The system performs other tasks in case of the absence of the necessary units for task $z_k$. When the unit becomes available, the model assigns them to the delayed task $z_k$.
3.　The model removes a unit from service in a defect or equipment failure, and replaces it with an available unit from the reserve $AU$.
4.　When a unit maintenance request is received, similar actions are taken.
5.　Continuous monitoring of the unit parameters "residual resource" and "time to preventive maintenance". When the specified resources are depleted, the unit is decommissioned and replaced with a reserve one.
6.　The model controls the unit queues for repair, maintenance, distribution, or recovery of consumable resources.
7.　Returning serviced units to reserve $AU$ for further use in the technological process.

A generalized hierarchical structure for scheduling production tasks, assigning units, and maintenance and repair processes is shown in Figure 1. The first level $L_1$ is formed by technological tasks performed according to the Gantt chart. The diagram shows units' movement from the state of availability to the state of activity (level $L_2$), as well as the transfer of units for repair or maintenance (level $L_3$). It is also possible to use the mobile service station $MM$, located closer to the technological process. The level $L_4$ corresponds to consumable resources $Cs$ and reusable resources $Ru$ for maintenance and repair.

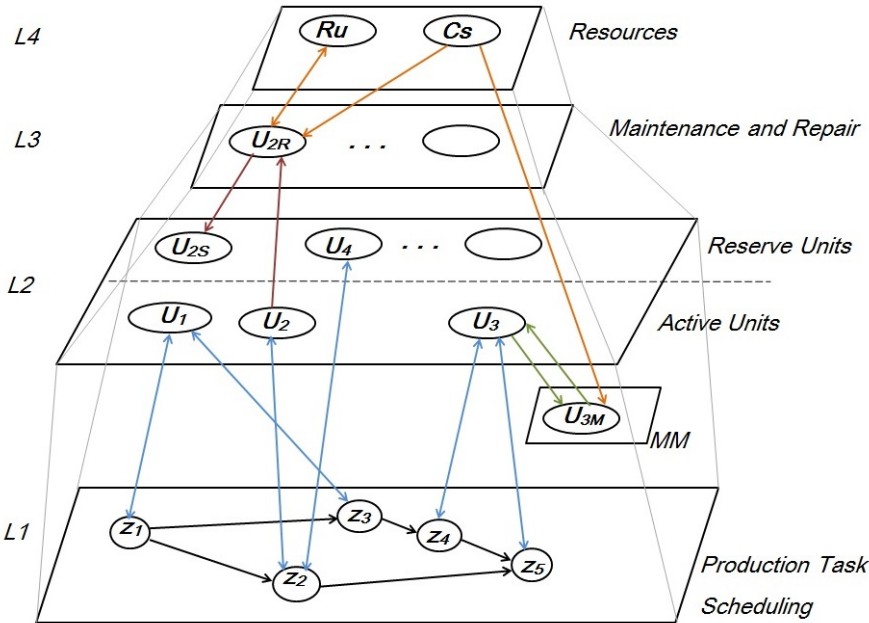

**Figure 1.** The scheme of interaction of technological objects and tasks of the technological process.

We describe the unit with a vector of parameters:

$$u = (ID, M_d, t_{rt}, t_{RL}, t_{MT}, t_{ti}, t_{ta}), \tag{1}$$

where $ID$ is the identification number of the unit, $M_d$ is the model (type) and modification of the unit, $t_{rt}$ is the value of the unit operating time after the last maintenance or repair procedure, $t_{RL}$ is remaining useful life (RUL), $t_{MT}$ is remaining time before scheduled maintenance, $t_{ti}$ is the start time of the next exploitation period, and $t_{ta}$ is the accumulated unit operating time.

Decision making is carried out according to the condition-based policy. The inspection time $t_I$ is related to the timing of the serviceability, degradation, and deterioration states of the unit components:

if $t_I \geq t_{RL}$ or $t_I \geq t_{MT}$ then: preventive replacement of unit;
if $t_I = t_{PrM}$ then: unit predictive replacement;
if $t_I = t_F$ then: unit corrective replacement;

where $t_{PrM}$ is the prognostic maintenance time, and $t_F$ is the time of a failure occurrence.

As follows from Figure 1, the modeled object has a hierarchical character, and belongs to the class of discrete-event systems. Therefore, the adequate model should also have a hierarchical structure. We have chosen hierarchical timed colored Petri nets to simulate the schedule, task execution events, maintenance, and unit repair. Figure 2 shows the relationship of the main modules of the hierarchical HPN model.

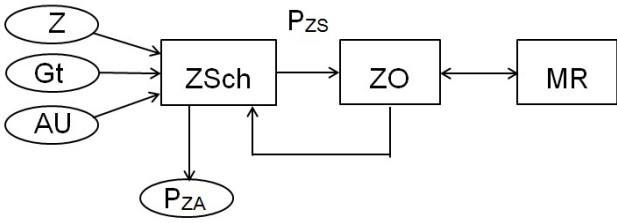

**Figure 2.** The main modules of the technological process hierarchical model.

The ZSch module, based on the specified Gantt chart Gt, builds a task graph and assigns unit types for each task. A P$_{ZS}$ set of assigned units are transferred to the ZO

module to complete tasks. It is possible that a technological task cannot be performed using the available set *AU* units. Such tasks $P_{ZA}$ are then removed from the model before decision making on the new unit's inclusion in the *AU*. During the execution of task Z, the following events may occur: (a) equipment failures; (b) requests for maintenance of units. Unit repair and maintenance operations are simulated in the MR module.

It is known that the coordination of task schedules with assigned technological equipment is the generalized assignment problem solution [48]. However, solving this problem poses difficulties for a large number of tasks and units. Furthermore, there are problems with the use of units in parallel tasks. We proposed to use an optimization–simulation approach to solve such problems [49,50]. This approach consists of the solution to the optimization problem and the study of the simulation model combined.

The optimization problem of unit allocation has the form:

$$F = \min\left\{ \sum_{n=1}^{N} C_n(x_n) + C_n^O(x_n) \right\}, \quad x_n \in \{0,1\}, n = \overline{1, N}, \tag{2}$$

under constraints:

$$\sum_n f_n(x_n) \leq 0, n \in N_j, j = \overline{1, J}, \tag{3}$$

where $x_n$ is the Boolean optimization variable, specifying the unit allocation $D(x_n)$, *J* is the number of constraints, *N* is the number of units, $N_j$ is an index set of units for the *J*th constraint, $|N_j| \leq N$, $C_n$ are capital costs of units, and $C_n^O$ are operating costs.

Variable $x_n = 1$ if the unit is assigned to execute a task in Z scheduling.

The problem shown in Equations (2) and (3) is not final, but serves the initial unit distribution by tasks in the simulation model. Simulation in HPNs is carried out with algorithmic restrictions. This allows the detection of situations such as deadlocks, resource insufficiency, and the simultaneous use of resources.

Thus, in general, decision making is an iterative process of solving an optimization problem, simulating, evaluating the results obtained, and returning to the optimization problem to correct the constraints if the algorithmic constraints are not satisfied.

## 4. Formal Description of Timed Colored PNs

We use a formal description of colored Petri nets as nonuples [16]:

$$\text{CPN} = (P, T, A, C, V, S, G, E, I), \tag{4}$$

where:

1.  P is a finite set of places.
2.  T is a finite set of transitions such that P∩T =/0.
3.  A ⊆ P × T ∪T × P is a set of directed arcs.
4.  C is a finite set of non-empty color sets.
5.  V is a finite set of typed variables such as Type [v] ∈ C for all variables v ∈ V.
6.  S: P→C is a color set function that assigns a color set to each place.
7.  G: T→EXPR is a guard function that assigns a guard to each transition such that Type [G (t)] = Bool.
8.  E: A→EXPR is a function that assigns an arc expression to each arc such that Type [E (a)] = C (p)$_{MS}$, where p is the place connected to the arc a.
9.  I: P→EXPR is an initialization function that assigns an initialization expression to each place p such that Type [I (p)] = C (p)$_{MS}$.

Timed colored Petri nets are:

$$\text{TCPN} = (\text{CPN}, \Theta_1, \Theta_2, \Theta_3), \tag{5}$$

where CPN stands for colored Petri net, and $\Theta_1$, $\Theta_2$, and $\Theta_3$ are time delays assigned to net places, arcs, and transitions, respectively.

We used CPN Tools 4.0.0 [51] to implement a hierarchical model of technological processes and equipment maintenance and repair. This software package's capabilities allow the multiset processing equipment to consider the timing relationship Gantt chart in order to analyze process control features.

## 5. Hierarchical Petri Net Model Used to Simulate the Repair and Maintenance Process

### 5.1. General Scheme

The design of the hierarchical HPN model was carried out using a top-down development. The top-level page uses subpages for which only a set of inputs and outputs are still defined. The structure of the hierarchical HPN model is depicted in Figure 3.

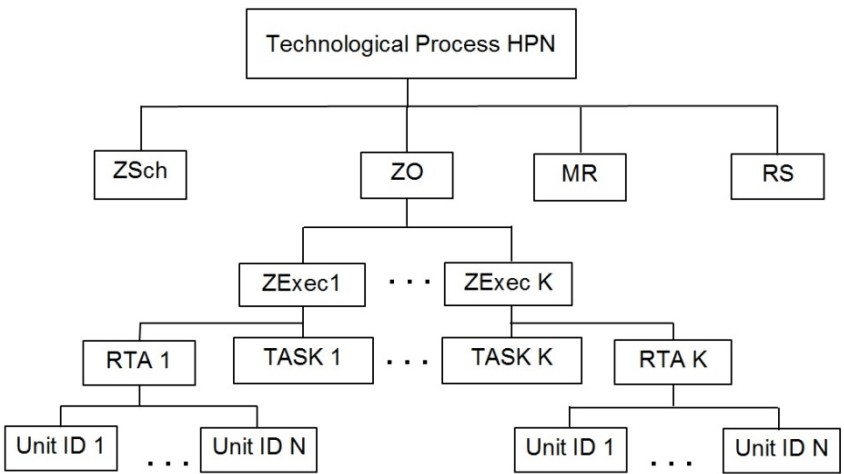

**Figure 3.** HPN structure.

Hierarchical PNs include four levels of modules. The first level consists of the following modules:

1. ZSch module handles task scheduling and unit assignment;
2. ZO module handles task operations;
3. MR module handles maintenance and repair performance;
4. RS module is the resource for maintenance and repair.

The first level is formed by the modules described in the previous paragraph. The main ZO module contains K basic ZExec modules of the task's operation, where K is the number of tasks in the technological process schedule. The ZExec module provides a simulation of unit assignment and basic task operations. Each ZExec module is associated with its own RTA (runtime analyzer) module, which evaluates the time parameters of running units and determines failures or predictive maintenance requests. The RTA module simulates an ensemble of m units for a specific task $z_k$. The Unit ID sub-module determines the accumulated operating time, remaining useful life, and time before preventive maintenance, and simulates stochastic processes of failures and prediction maintenance. A detailed description of the implementation of HPN modules is given below.

### 5.2. ZO Module

All basic operations for performing production tasks are simulated in the ZO module. The topology of the ZO module is determined by the process Gantt diagram. Each task has a base operations module BOM associated with it. Figure 4 shows the ZO module structure as an example, including three blocks of operations BOM1–BOM3; these are substitution transitions that correspond to ZExec modules on the lower level of the model hierarchy.

ZBM modules are intermediate substitution transitions to ZExec modules. The example (Figure 4) implements sequentially parallel execution of the production "Main Task."

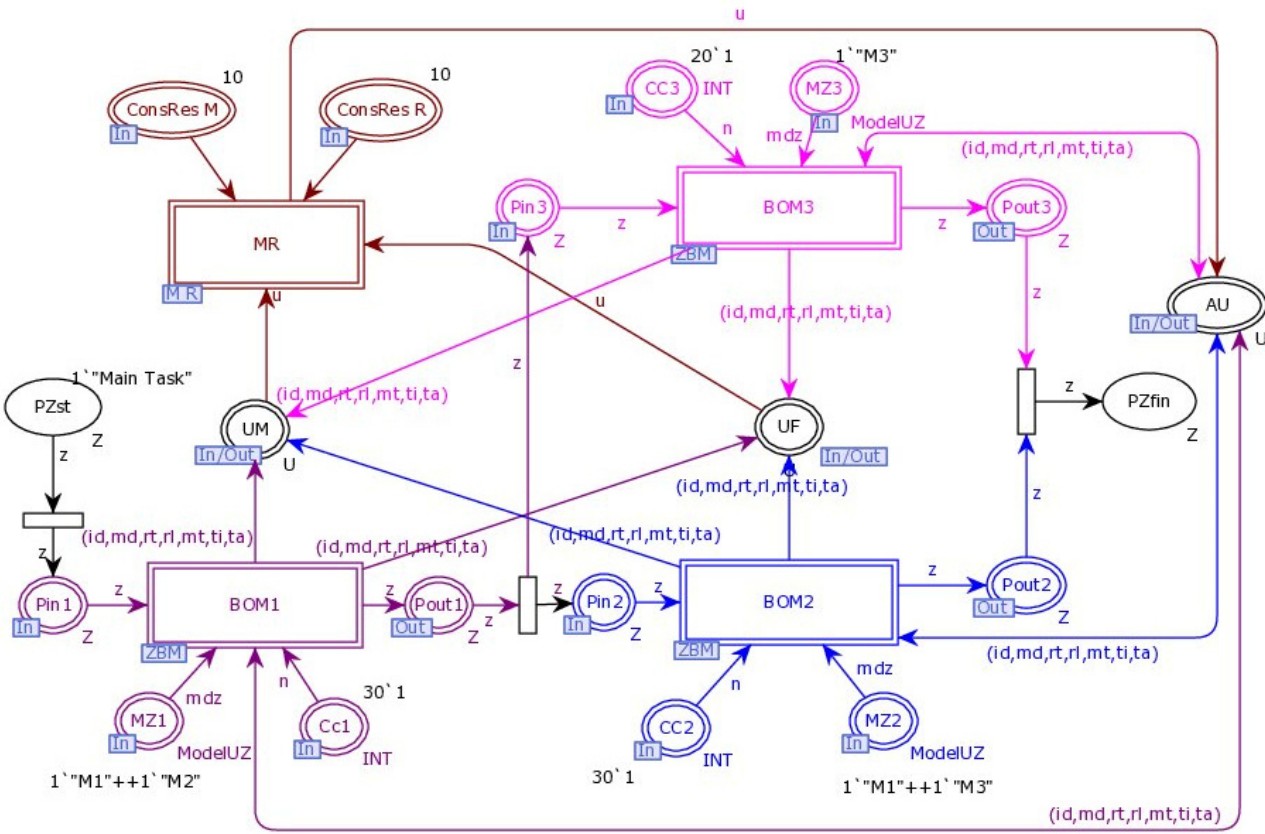

**Figure 4.** ZO module.

The global color sets and variables are given in Table 1. Table 2 contains a description of the ZO places.

**Table 1.** Global color sets.

| Color Set | Color Variables | Meaning |
|---|---|---|
| ID | id | Unit identification number |
| Model | md | Model of a unit |
| RT | rt | Unit's operating time $t_{rt}$ |
| RL | rl | Unit's remaining useful life $t_{RL}$ |
| MT | mt | The unit's remaining time before maintenance $t_{MT}$ |
| TI | ti | Unit's exploitation start time |
| TA | ta | Unit's accumulated operating time $t_{ta}$ |
| U = product ID*Model*RT*RL*MT*TI*TA | u, upd_u | Parameters of a Unit |
| Z | z | Task |
| BOOL | b | Failures and maintenance request events |
| ModelZ | mdz | Models of the units assigned to a task |

Thus, according to Equation (1), the main color set U is defined in the HPN model as a tuple U = (id, md, rt, rl, mt, ti, ta). The unit's unique id and model md associate it with the tasks being performed. The time parameters rt, rl, mt, ti, and ta are used to control the unit's residual life. Only the active unit of work time determines the remaining useful life RUL (rl parameter). In periods of service or repairs to the unit, time rt does not change.

**Table 2.** ZO module places.

| Place | Meaning | Color Set |
|---|---|---|
| PZst | Input place of ZO module | Z |
| PZfin | Output place of ZO module | Z |
| Pin | Input place of ZBM module | Z |
| Pout | Output place of ZBM module | Z |
| AU | The tokens represent available units | U |
| UPZ 1,..., UPZ K | The tokens represent assigned units for the task | U |
| UM, UF | The tokens represent the units in a maintenance queue or repair queue | U |
| MZ 1,..., MZ K | The tokens represent the unit models assigned to the task's operations | ModelUZ |
| CC1,..., CCK | The tokens represent task cycle numbers | INT |
| ConsRes M, ConsRes R | The tokens represent the consumable resources | INT |

Then from Equation (1), the expression for the unit current operating time $t_{cu}$ follows:

$$t_{cu} = t_c - t_{ti} + t_{rt}, \tag{6}$$

where $t_c$ is the model time.

Requests for the unit's predictive maintenance are simulated in the process of performing tasks. For this purpose, the random distribution functions are used corresponding to the degradation and deterioration processes of unit parts. At random time $t_{Mrq}$, the Mrq signal is formed, causing a transfer of the unit for maintenance. Similarly, at the moment of time $t_F$, the signal F is generated to simulate the failure occurrence.

The unit is operated in the technological process under the following conditions:

- The inspection time is less than the value of RUL defined for each unit.
- The inspection time is less than the time before scheduled maintenance.
- There are no signals about the appearance of failure.
- There are no predictive maintenance requests.

Therefore, the step sequence for simulating a technological process execution is as follows:

1. Module initialization: Unit model types that can perform operations on the BOM module are set in place MZ. The number of operation cycles is set in place Cc. If necessary, the FE (failure enable) and PrME (predictive maintenance enable) indicators are set.

2. Each $BOM_k$ module assigns specific units to perform task $z_k$ according to the rule:

$$\exists u : u_n \in AU \wedge md(u_n) = mdz(BOM_k). \tag{7}$$

As a result, the places UPZ 1–UPZ K contain sets of unit models that can perform these tasks throughout the entire period. Note that the set $AU$ of available units can contain several units of the same model with different $ID$s.

3. The "Main Task" token movement through the BOMs simulates the execution of a technological process. When a task is completed in the BOM, the place EndZ in the ZExec module receives a token.

4. The RTA analyzer checks the unit operating conditions throughout the BOM module's activity period. If conditions 1 and 2 are not satisfied, the residual resource is exhausted; then, the unit stops working, and it is directed to preventive maintenance.

5. In cases where the FE and PrME features are set, the RTA analyzer generates random failure or defect prediction events; then, at the appropriate time, the unit is transferred for repair or maintenance. This action is simulated by moving the unit token to the places UM or UF, and then to the MR module, for maintenance or repair operations. During this period, the unit token is only in the MR module, and the value of the operating time $t_{rt}$ does not change.

6. If the task is suspended, then another unit with the required model is selected from the set $AU$ to the BOM module, and the task continues.

7. The "Main Task" token appears at the output place PZfin when the technological process is completed.

### 5.3. ZExec Modules

The ZExec module is a substitution transition for ZBM modules. Figure 5 shows the structure of a ZExec with substitution transitions TASK and RTA.

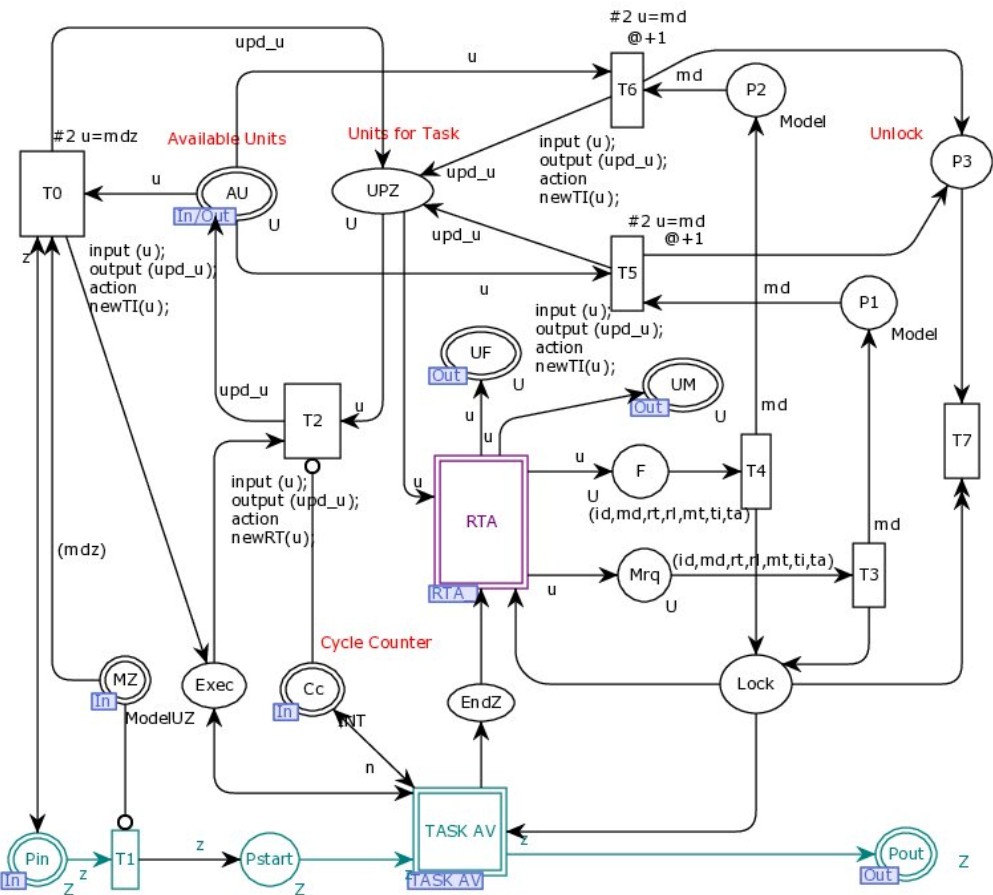

**Figure 5.** Structure of the ZExec module.

The appearance of a token at the input position Pin allows the assignment of units to execute the task. A set of required unit models is contained in the place MZ. For the transition T0, a code segment is defined that specifies the function newTI (u) for setting the start time $t_{ti}$ for a given operating period. The task operations begin after moving units from place *AU* to place UPZ.

The operation of module ZExec when simulating failures or predictive maintenance requests is as follows:

1. The token appearance in F or Mrq places determines the model of the new unit that should replace the unit being serviced or repaired. Transitions T3 and T4 send the lock token to the place Lock.
2. The T5 or T6 transitions fire if a unit with a required model is in place *AU*; then, the unit is replaced in UPZ.
3. The transition T7 unlocks the operation execution after replacing a unit in place UPZ.
4. At the end of the task, the Cc counter is reset to zero, and the inhibitor arc starts transition T2 to move units from place UPZ to place *AU*, to be used in other BOM modules. The function newRT (u) sets new values of $t_{rt}$ and $t_{ta}$.

In the ZExec module, the meaning of places and transitions are explained in Tables 3 and A1.

**Table 3.** Transitions of ZExec module.

| Transitions | Description of Transitions |
|---|---|
| T0 | Selecting a unit from the general set of available units |
| T1 | Start task execution |
| T2 | Returning units to AU after completing a task |
| T3, T4 | Selecting the unit model for replacement |
| T5, T6 | Replacing units in UPZ |
| T7 | Unlocking task execution |

### 5.4. TASK Sub-Module

The hierarchical approach assumes that the internal organization of any technological task can have a complex structure. Figure 6 shows a sub-module with three phases of task execution.

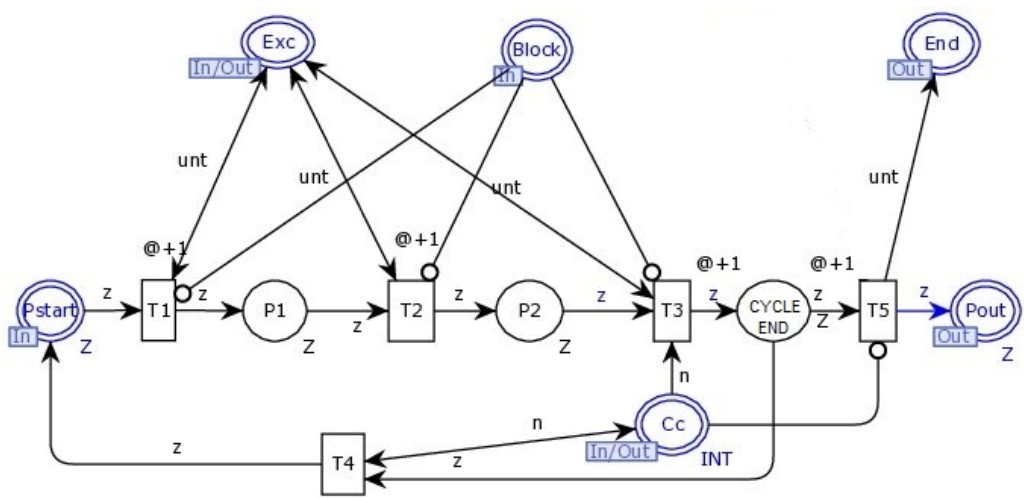

**Figure 6.** The three-phase TASK sub-module.

The transition T1 performs a preparatory operation for the task execution. The transition T2 corresponds to the main operation, and the firing of the transition T3 is the final action. The iterative cycle of operations is organized by transitions T3 and T4 and places Cc and CYCLE END. After all of the task operations, the units are removed from the place UPZ in the ZExec module. An empty token in place CYCLE END allows the firing of transition T5 through the inhibitor arc. It sets the flag End and moves the token "Main Task" to the place Pout. This concludes the work of the TASK and ZExec modules.

Such a task structure is the simplest case, but it is often encountered in applications. Next, an example of the practical use of a simulation model will be considered, while the TASK module has a more complex structure.

### 5.5. RTA Module and Unit ID Sub-Module

Each RTA is assigned to a single task $z_k$, and is associated with the corresponding BOM module. Essentially, RTA is a multifunctional unit's time counter, and includes several Unit ID sub-modules (Figure 3). The structure of the Unit ID module is depicted in Figure A1. The meanings of the module's places and transitions are briefly explained in Tables A2 and A3.

The primary operations of the Unit ID module are:

- Selecting a unit with a given ID = j from place UPZ and moving to place UPZ j;
- Setting in place E the status "Unit activity" and starting the internal time counter;
- Checking the conditions $n \geq rl$ (RUL) and $n \geq mt$ (time before maintenance) performed on the output arcs of the transitions T2 and T3 in each time step;

- If at least one condition is satisfied, the unit is transferred to the MR module by transition Tout2;
- Generating a predictive maintenance request on the transition T4 output arc according to the Poisson distribution, with a given rate $\lambda_M$, and moving the unit to the place UM;
- Generating a failure event on the transition T5 output arc according to the Poisson distribution, with a given rate $\lambda_F$, and moving the unit to the place UF;
- Removing the unit from the place UPZ j, if it becomes inactive;
- Setting new values of unit accumulated operating time $t_{ta}$ and $t_{rt} = 0$ using the code segments of function newRTLM (u);
- Resetting the module marking to its initial state upon work completion on a signal in the place EndZ.

The researcher can study the influence of stochastic processes in the simulated equipment. Tokens in the FE and PrME positions of the Unit ID module allow the firing of transitions T4 and T5. The output arc expressions for these transitions generate random times of failure or requests from the predictive maintenance system. CPN Tools provides random distribution functions: normal, Poisson, Weibull, gamma, exponential, and others. The example module in Figure A1 shows Poisson failure flow generation with a rate of $\lambda = 3$. It should be noted that the values of the random distribution parameters depend on the ratio of the model and the real time of the technological process.

### 5.6. Maintenance and Repair Module

Repairs and maintenance are simulated in the MR module (Figure 7). The queues of units for maintenance and repair are formed in the places UMQ and UFQ, respectively. Tokens in these places are transferred from the ZExec modules. The tokens in the places AM and AR allow the maintenance or repair operations for the next unit. At the same time, tokens leave these places through transitions T1 and T2, blocking the service of new units. Basic manufacturing operations are simulated in the M and R transitions. The presence of the required number of consumable resources is indicated in the places CsM and CsR. The restored unit is moved to the place *AU* in the available unit set. Tables 4 and 5 describe the meaning of MR module places and transitions.

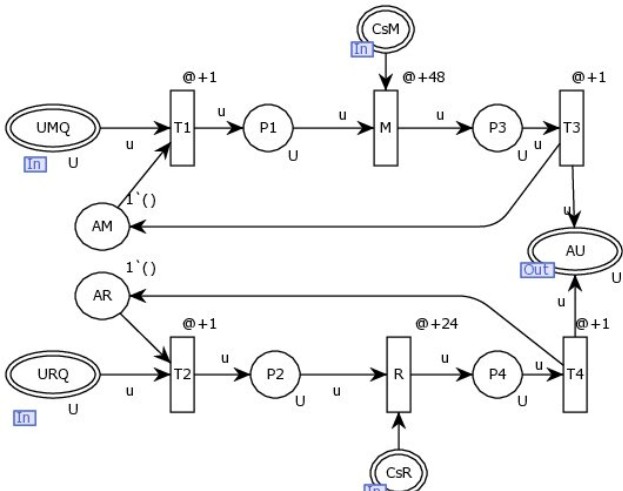

**Figure 7.** MR module.

**Table 4.** Places of MR module.

| Place | Meaning | Color Set |
|---|---|---|
| UMQ | The tokens represent the units in a queue to maintenance | U |
| URQ | The tokens represent the units in a queue to repair | U |
| AU | Set of available units | U |
| P1, P2 | The tokens represent units under maintenance or to be repaired, respectvely | U |
| P3, P4 | The tokens represent units after maintenance or repair, respectvely | U |
| AM, AR | Maintenance or repair activation | UNIT |
| CsM, CsR | The tokens represent the consumable resources for maintenance and repair, respectively | INT |

**Table 5.** Transitions of MR module.

| Transitions | Description of Transitions |
|---|---|
| T1, T2 | Preparing for maintenance or repair |
| M | Performing maintenance |
| R | Performing repair |
| T3, T4 | Returning a unit to the set of available units |

## 6. Experiments and Results

### 6.1. Simulation of Autonomous Agricultural Vehicles (AAVs)

The robotic system of autonomous agricultural vehicles (SAAV), based on the KA-MAZ vehicle family, with autonomous and remote control, is a human–machine system. The system aims to perform transport tasks using a KAMAZ robotic chassis controlled by a dispatching subsystem. A robotic car is a cyber–physical system; its functioning is based on a combination of complex physical and information processes: the physics of fuel combustion, mechanical motion, material fatigue processes, information processes, intelligent data processing systems, and motion control using machine vision [52]. The construction of full-scale models in the design of such systems is costly, and does not allow for the consideration of various scenarios or prediction of the AAV's performance. In this regard, an effective method is to integrate digital models of AAV aggregates and subsystems and conduct numerous digital tests [53].

In a predictive maintenance system, the parameters characterizing the actual state of the AAV come from the onboard system. These are periodically compared with the AAV models' parameters, taking into account the wear of components and environmental influences. The AAV's state is assessed by the deviation magnitudes of the parameters from the model values. The alert states of the subsystems and units are determined, and the residual life of the equipment and subsystems is forecast. Based on the analysis, the predictive maintenance system generates recommendations for the AAV service personnel regarding the timing of component replacement and the maintenance schedule.

SAAV consists of the following subsystems (Figure 8):

- AAV group;
- Data transfer system;
- AAV system of remote diagnostics and predictive maintenance;
- AAV models (digital twins).

### 6.2. Hierarchical Petri Net Model for Autonomous Vehicle Simulation

The simulation experiment to study the predictive maintenance of the AAV is conducted using a model similar to a hierarchical HPN model (Figure 3). The difference between the models is the implementation of the TASK module. It is assumed that the technological process of delivering goods to/from the location of the agrotechnical production

task includes five operations. The TASK AV module structure for this case is depicted in Figure 9.

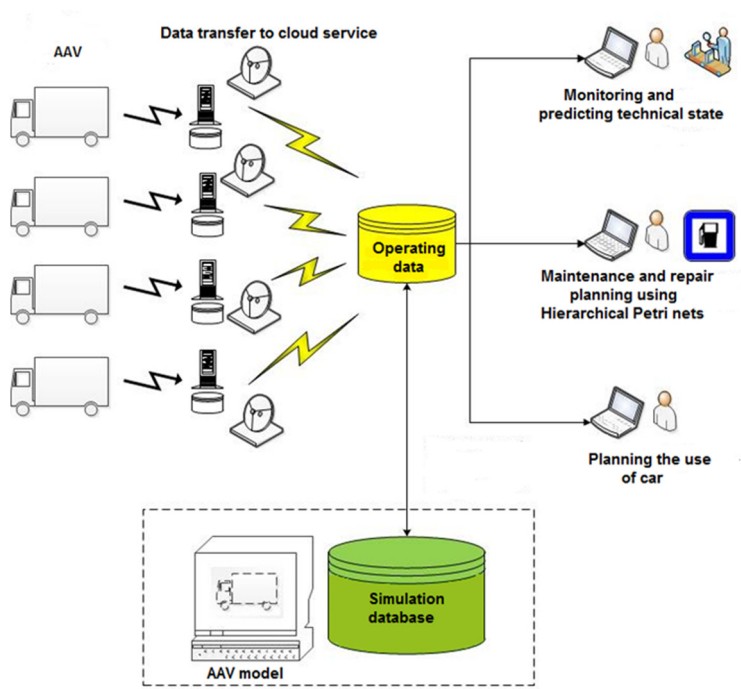

**Figure 8.** Predictive maintenance system for autonomous agricultural vehicles.

The production task consists of operations that are simulated by the following transitions:

- Transition T1 is the AAV's movement along a given route from the home base to the operational position;
- Transitions T2 and T4 are maneuvering near the operating position;
- Transition T3 is the execution of the main production operation;
- Transition T5 is the AAV's return route to the place of deployment.

It should be noted that the processes of vehicle maneuvering in the field when approaching the target place, in many cases, take place under increased loads and increased risk of emergencies. The TASK AV module simulates these factors.

The HPN AV model structure description provided in the CPN Tools, and the HPN declarations, are shown in Figures A2–A4.

An example of evaluating the AAVs' predictive maintenance effectiveness is made for a parallel–sequential schedule of three tasks and three vehicles. It simulates the KAMAZ three models, which correspond to the value of the variable md for color set U:

$$\text{md} = \text{"M1"} \rightarrow \text{KAMAZ} - 43{,}502 - \text{lorry with sides;}$$

$$\text{md} = \text{"M2"} \rightarrow \text{KAMAZ} - 45{,}143 - \text{dump truck;}$$

$$\text{md} = \text{"M3"} \rightarrow \text{KAMAZ} - 65{,}111 - \text{dump truck.}$$

Figure A3 shows the initial marking of the ZO module, with vehicle models M1 and M2 assigned to BOM1, models M1 and M3 to BOM2, and model M3 to BOM3. Therefore, the initial marking of the ZO module places is represented by multisets:

$$\text{MZ1} = 1'\,\text{"M1"} ++ 1'\,\text{"M2 "}; \text{MZ2} = 1'\,\text{"M1"}++ 1'\,\text{"M3"}; \text{MZ3} = 1'\,\text{"M3"}.$$

For this example, the final marking of HPN AV is depicted in Figure A4. Studies of KAMAZ family vehicles were carried out in order to assess the degradation and deterioration of their components and assemblies. Tables 6 and 7 list MTBF (mean time between failures)

data for the autonomous vehicles' brake systems and fuel systems. Two distributions are involved here: normal, with mean μ and standard deviation σ; and Poisson, with failure rate λ per 10,000 h.

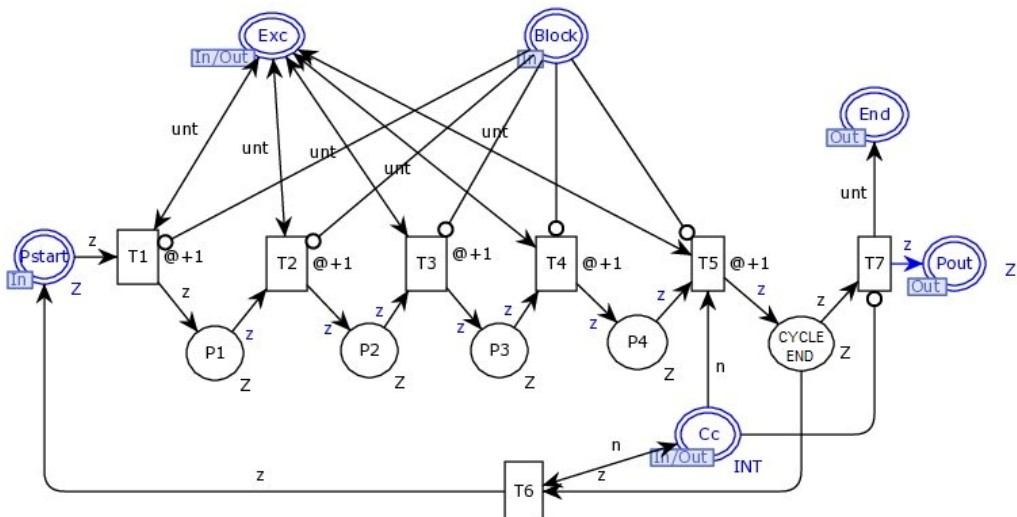

**Figure 9.** TASK AV module with five phases of operations for autonomous agrotechnical vehicles.

**Table 6.** Brake system.

| Component | Failure Time Distribution | Values, h |
|---|---|---|
| Front axle modulator solenoid | Normal | μ = 11,675; σ = 587 |
| Rear and center axle solenoid | Normal | μ = 10,900; σ = 1100 |
| Brake drum | Normal | μ = 4140; σ = 295 |
| Brake pad | Normal | μ = 2032; σ = 172 |
| Pneumatic cylinder | Normal | μ = 11,080; σ = 1002 |
| Air compressor bleed valve | Poisson | λ = 0.4 |
| Air compressor gear | Poisson | λ = 0.4 |
| Air compressor piston | Poisson | λ = 0.38 |

**Table 7.** Fuel system.

| Component | Failure Time Distribution | Value |
|---|---|---|
| Fuel nozzle | Poisson | λ = 0.38 |
| Fuel pump | Normal | μ = 6170; σ = 535 |
| Fine fuel filter | Poisson | λ = 0.32 |
| Fuel strainer | Poisson | λ = 0.46 |
| Fuel pressure sensor | Normal | μ = 5583; σ = 513 |
| Fuel temperature sensor | Normal | μ = 5760; σ = 540 |
| Fuel pump drive | Normal | μ = 9200; σ = 750 |

Analyzing the time scales of service processes and the occurrence of failures in KA-MAZ vehicles, the following relationship between model time (MTU) and real time was adopted: 1 MTU = 1 h.

To obtain the integrated parameters of the car from the data of its components, we used the technique described in [54]. A mathematical model for determining the uptime probability was based on monitoring the vehicle mileage, the impact of factor influence, and the residual component life. As a result, for the considered example of modeling the fuel and braking systems, the integral stochastic parameters of failures were obtained. For example, Table 8 shows the time parameters of three AAVs given in the model time. At the initial time taken RT = 0 and TI = 0.

**Table 8.** AAV unit parameters.

| ID | Model | RT | RL | MT | TI | TA |
|----|-------|-----|------|------|----|----|
| 1 | M1 | 0 | 2000 | 400 | 0 | 0 |
| 2 | M2 | 0 | 5000 | 1400 | 0 | 0 |
| 3 | M3 | 0 | 4000 | 5500 | 0 | 0 |

We evaluate the maintainability effectiveness by the schedule completion degree of the technological process. Factor $Y$ is determined in the form of a task completion delay ratio obtained during statistical tests by the Monte Carlo method:

$$Y = \frac{\mu(t_A) - t_Z}{t_Z} \times 100\%,$$

where $t_Z$ is the task's scheduled completion time, $t_A$ is the task's actual completion time, and $\mu(t_A)$ is the average actual completion time.

For determining the required set of vehicles for the technological process, the AAV operating rate $E_j$ is applied:

$$E_n = \frac{\sum\limits_{k} t_{jk}}{\sum\limits_{k} t_{zk}} \times 100\%, k \in I_j,$$

where $t_{zk}$ is the task $z_k$ execution time, $t_{jk}$ is the operation time for AAV with id = $j$ during task $z_k$ execution, and $I_j$ is the index set of tasks to which $j$th car is assigned.

The HPN model has been tested on multiple datasets corresponding to various repair and maintenance modes for autonomous vehicles. In the presented article, the results of the following experiment are given as an example: statistical tests were conducted under the normal distribution of the AAV maintenance time at the service station and a set of time values until preventive maintenance.

Tables A4 and A5 contain the values of parameters of the hierarchical PN model, and show obtained factor $Y$ and operating rate $E$. In addition, the charts of factor $Y$ and operating rate $E$ are presented in Figures 10 and 11, illustrating the parameter interdependencies.

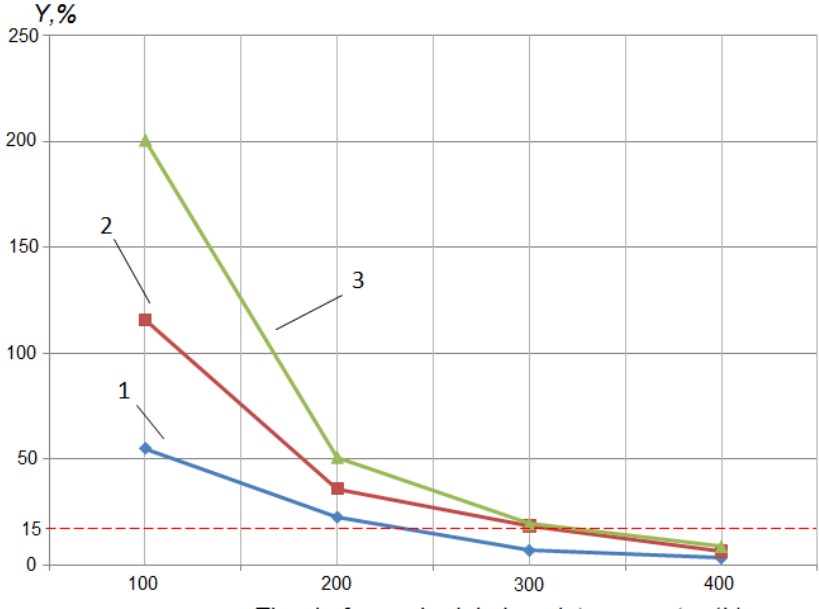

**Figure 10.** Task completion delay ratio $Y$: (**1**) $t_M = 24$ h; (**2**) $t_M = 48$ h; (**3**) $t_M = 72$ h.

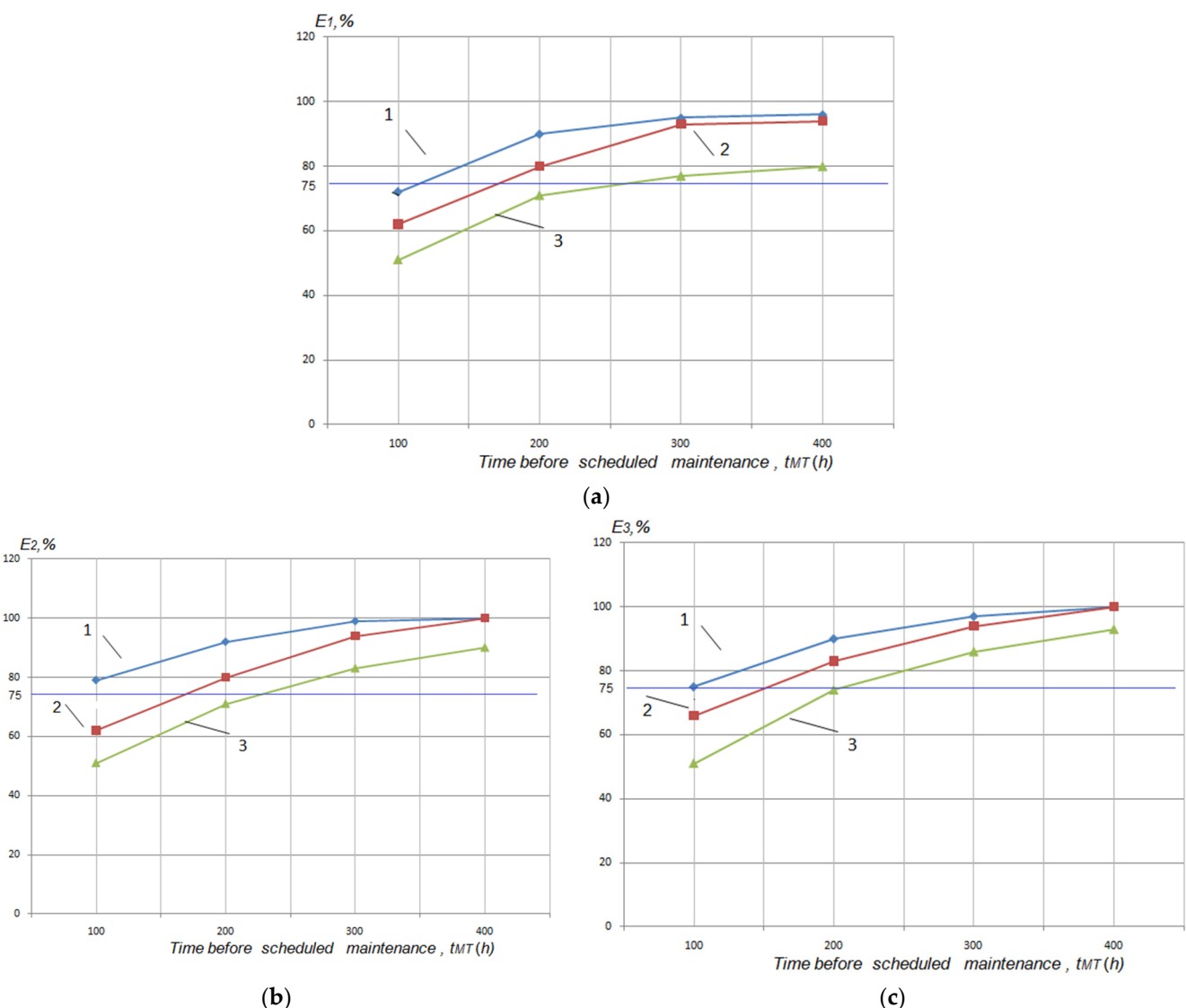

**Figure 11.** The parameters of AAV system: (**a**) Operating rate $E_1$; (**b**) Operating rate $E_2$; (**c**) Operating rate $E_3$; (**1**) $t_M$ = 24 h; (**2**) $t_M$ = 48 h; (**3**) $t_M$ = 72 h.

## 7. Discussion

The developed hierarchical CPN model describes many objects, their states, and their parameters. The model allows us to study various properties in the autonomous vehicle system, and make decisions to improve process control. Planning multifactorial experiments is based on the identification of the main components that affect key indicators. Based on the results of the presented experiment, the following conclusions can be drawn:

The researcher sets the limit value for exceeding the scheduled completion time of the technological task Z. Assume that max $Y$ = 15% (red dotted line in Figure 11). The graphs in Figure 11 show that for $Y \leq 15\%$, the acceptable value range for the parameters $t_{MT}$ and $t_M$ is determined by the inequalities:

$$t_{MT} \geq 230 \text{ h}, \ t_M = 24 \text{ h};$$
$$t_{MT} \geq 310 \text{ h}, \ t_M = 48 \text{ h};$$
$$t_{MT} \geq 320 \text{ h}, \ t_M = 72 \text{ h}.$$

Hence, the preventive maintenance period for autonomous vehicles must be at least 320 h. Otherwise, the AAV's transfer to service and repair work leads to a significant

slowdown in the technological process. The useful time of autonomous vehicles is similarly estimated using the operating rate indicator $E$. For a given rate $E \geq 75\%$, the parameters $t_{MT}$ and $t_M$ must satisfy the following ratios:

$$t_{MT} \geq 120 \text{ h}, \ t_M = 24 \text{ h};$$
$$t_{MT} \geq 175 \text{ h}, \ t_M = 48 \text{ h};$$
$$t_{MT} \geq 280 \text{ h}, \ t_M = 72 \text{ h}.$$

By combining the conditions Y and E, we obtain the limit value $t_{MT} = 320$ h.

We are also considering using reserve autonomous vehicles to reduce the loss of repair and maintenance time. To do this, in HPN AV, a multiset of place *AU* is added, according to a fourth vehicle with model "M1" and id = 7. Thus, when a car with id = 1 is brought to the service, it is replaced by a reserve car with id = 7 when performing tasks $z_1$ and $z_2$. The experiment shows that, for $t_M = 48$ h and $t_{MT} = 200$ h, the *Y* criterion decreases from 67% to 30%. At the same time, the vehicle rate *E* also decreases (Table 9). However, this leads to the timely completion of task *Z*.

**Table 9.** AAV reserve efficiency assessment.

| Maintenance Time $t_M$, h | | Time Before Scheduled Maintenance $t_{MT}$, h | Operating Rate $E_1$ for AAV1, % | Operating Rate $E_2$ for AAV2, % | Operating Rate $E_3$ for AAV3, % | Operating Rate $E_4$ for AAV4, % |
|---|---|---|---|---|---|---|
| | A [1] | 100 | 62 | 63 | 66 | - |
| $\mu = 48$; | B [2] | 100 | 51 | 59 | 58 | 40 |
| $\sigma = 6$ | A [1] | 200 | 80 | 80 | 80 | - |
| | B [2] | 200 | 37 | 86 | 64 | 62 |

[1] Three autonomous vehicles; [2] three autonomous vehicles and reserve AAV4.

These results are taken into account when correcting the constraints of the optimization problem shown in Equations (2) and (3) to find a balance between the unproductive idle time of vehicles and the degree of compliance with the schedule.

## 8. Conclusions

The hierarchical HPN model presented in this article can describe processes in a comprehensive class of flexible manufacturing systems. However, some structural and technical features of this model determine its most effective area of application. In the HPN model, the emphasis is on putting units into operation, their replacement, and the formation and use of a reserve of units. In this regard, the model is more suitable for the organization of maintenance and repair of mobile industrial equipment.

It should be noted that the novelty of our approach also lies in the application of an optimization–simulation procedure. In modeling the maintenance of the autonomous robotic vehicle system, we managed to solve the problem of the rational placement of AAVs in a production area with the required number of reserve units. The simulation experiment results obtained via the Monte Carlo method were used to adjust the constraints of minimizing the maintenance cost, as shown in Equations (2) and (3). We can say that such an iterative procedure is, to some extent, analogous to machine learning; the difference is that, at this stage of our work, the decision to correct the optimization problem is made by the production system's analyst/manager. In our study of AAV systems, we obtained solutions close to optimal, in the performance of no more than 15–20 iterations.

The HPN model contains a set of UNIT ID modules, which are individual multi-functional timers for each technology unit. This solution is new, and allows us to keep track of the time parameters of the unit's activity, the residual resource, and the remaining time before maintenance. The UNIT ID module performs these functions regardless of the production task or the place of the unit's use. This provides the ability to quickly move information about a unit to other hierarchical models, in order to transfer it to a new production system.

It should be noted that the structure of the hierarchical model and its main modules can be adapted to a vast class of flexible production systems. The ZO module and the ZExec module structure allow the substitution of local models of various production tasks and technological equipment units. However, there are certain limitations to the application of the proposed hierarchical model.

First, the desire to perform deeper detailing, in order to study the stochastic processes of the behavior of individual subsystems and elements of technological units, leads to congestion of the model's connections and an increase in the dimension of the PN model. As a result, the simulation time increases significantly, and the analysis of the obtained solutions becomes more complicated.

Secondly, the tools that support PN modeling have limitations on the number of places and transitions in the model. Experiments with the AAV system have shown that it makes no sense to build a complete simulation model for an enterprise with 50–100 robotic vehicles. It is enough to divide the AAV fleet into groups that perform local production tasks, with the number of vehicles no more than 15. In this case, the simulation can be performed reasonably, and the obtained solution adequately describes the manufacturing system.

A possible direction for further research is the development and complication of the hierarchical model of autonomous agricultural vehicles. The idea is that AAVs include several microcontroller-based control subsystems. Predictive maintenance uses information obtained from monitoring while operating the vehicle in challenging road conditions. Therefore, it is necessary to perform the detailing of the HPNs at the lower levels of the model. This approach allows the study of process failures in microcontrollers, measurement subsystems, and data transmission devices. In this case, external signals and factors should be investigated. Hence, the model HPN AV is transformed into a non-autonomous Petri net. Some works are known in the field of the application of non-autonomous Petri nets. L. Gomes et al. conducted research in which they implemented IOPT models for simulating microcontrollers [55–57]. There are many reasons to use the IOPT tool [58]; in this case, the basic premise is related to describing interfaces and signals in the onboard diagnostic system of an autonomous vehicle.

**Author Contributions:** Conceptualization, S.P.O. and S.V.S.; methodology, S.V.S.; software, R.A.U.; validation, S.V.S. and R.A.U.; writing—original draft preparation, S.V.S.; writing—review and editing, S.P.O.; visualization, R.A.U.; supervision, S.P.O.; project administration, S.V.S. All authors have read and agreed to the published version of the manuscript.

**Funding:** This research was funded by Russian Foundation for Basic Research, grant number 20-37-90014.

**Institutional Review Board Statement:** Not applicable.

**Informed Consent Statement:** Not applicable.

**Data Availability Statement:** No new data were created in this study. Data sharing is not applicable to this article.

**Conflicts of Interest:** The authors declare no conflict of interest.

# Appendix A

**Table A1.** Places of module ZExec.

| Place | Meaning | Color Set |
|---|---|---|
| $P_{in}$ | Input place | Z |
| $P_{out}$ | Output place | Z |
| $P_{Start}$ | The token represents the task operation start | Z |
| UPZ | The tokens represent units that perform the task | U |
| AU | The tokens represent an available unit | U |
| MZ | The tokens represent unit models suitable for task operations. | Model |
| P1, P2 | The tokens represent the unit models to replace at UPZ | Model |
| P3 | Auxiliary place | UNIT |
| Cc | The tokens represent a task cycle number | INT |
| F, Mrq | A token represents the type of model for replacing a unit in position UPZ | U |
| Exec | A token represents a task execution enable | Unit |
| Lock | A token represents blocking the execution | Unit |
| EndZ | A token represents the task execution finish | Unit |

**Table A2.** Places of Unit ID modules.

| Place | Meaning | Color Set |
|---|---|---|
| UPZ j | A token represents a unit with ID = j | U |
| ID | Unit identifier | INT |
| E | Enable analysis of time stamps | UNIT |
| time | Model current time counter | INT |
| Time1, Time2 | Current time for condition check | INT |
| m | ID of the unit for maintenance | INT |
| FE | The occurrence of a failure | BOOL |
| f | ID of the unit for repair | INT |
| PrMAe | The occurrence of a predictive maintenance request | BOOL |
| UM, UF, F, Mrq | A token represents a unit for replacing | U |
| EndZ | Task completion and reset | UNIT |

**Table A3.** Transitions in the RTA module.

| Transitions | Description of Transitions |
|---|---|
| Tin | Input unit data $u_j$ from a place UPZ |
| T1 | Forming the current module time |
| T2, T3 | Checking unit stop conditions |
| T4, T5 | Generating random timestamps |
| T6, T7 | Permission to output unit data to the MR module |
| Tout1 | Output unit $u_j$ for maintenance |
| Tout2 | Output unit $u_j$ for repair |

**Table A4.** Maintenance efficiency assessment.

| Maintenance Time $t_M$, h | Time Before Scheduled Maintenance $t_{MT}$, h | Average Time $\mu(t_A)$, h | Task Completion Delay Ratio $Y$, % |
|---|---|---|---|
| M = 24; σ = 3 | 100 | 1396 | 55 |
| | 200 | 979 | 22.7 |
| | 300 | 885 | 7.1 |
| | 400 | 831 | 3.5 |
| M = 48; σ = 6 | 100 | 1773 | 116 |
| | 200 | 1094 | 36 |
| | 300 | 948 | 18.6 |
| | 400 | 852 | 6.5 |
| M = 72; σ = 8 | 100 | 2412 | 201 |
| | 200 | 1214 | 51 |
| | 300 | 939 | 17.3 |
| | 400 | 872 | 8.8 |

**Table A5.** AAV use assessment.

| Maintenance Time $t_M$, h | Time Before Scheduled Maintenance $t_{MT}$, h | Operating Rate $E_1$ for AAV1, % | Operating Rate $E_2$ for AAV2, % | Operating Rate $E_3$ for AAV3, % |
|---|---|---|---|---|
| M = 24; σ = 3 | 100 | 72 | 79 | 75 |
| | 200 | 90 | 92 | 90 |
| | 300 | 95 | 99 | 97 |
| | 400 | 96 | 100 | 100 |
| M = 48; σ = 6 | 100 | 62 | 63 | 66 |
| | 200 | 80 | 80 | 80 |
| | 300 | 93 | 94 | 93 |
| | 400 | 94 | 100 | 100 |
| M = 72; σ = 8 | 100 | 51 | 58 | 51 |
| | 200 | 71 | 71 | 74 |
| | 300 | 77 | 83 | 86 |
| | 400 | 80 | 90 | 93 |

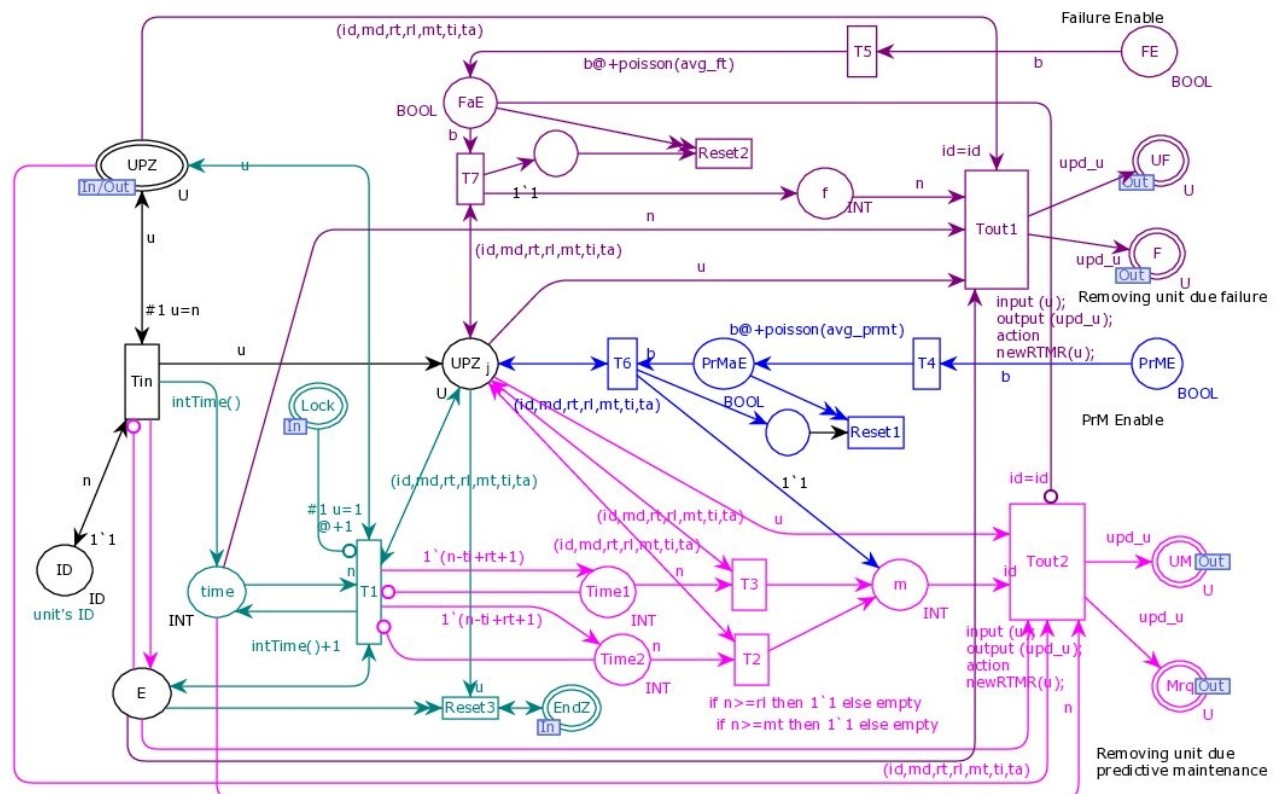

**Figure A1.** UNIT ID module.

Figures A2–A4 describe the HPN AV model's description corresponding to the example in Section 6.2.

(**a**)　　　　　　　　　　　　　　　　(**b**)

**Figure A2.** HPN AV model in CPN Tools: (**a**) hierarchical structure; (**b**) declarations.

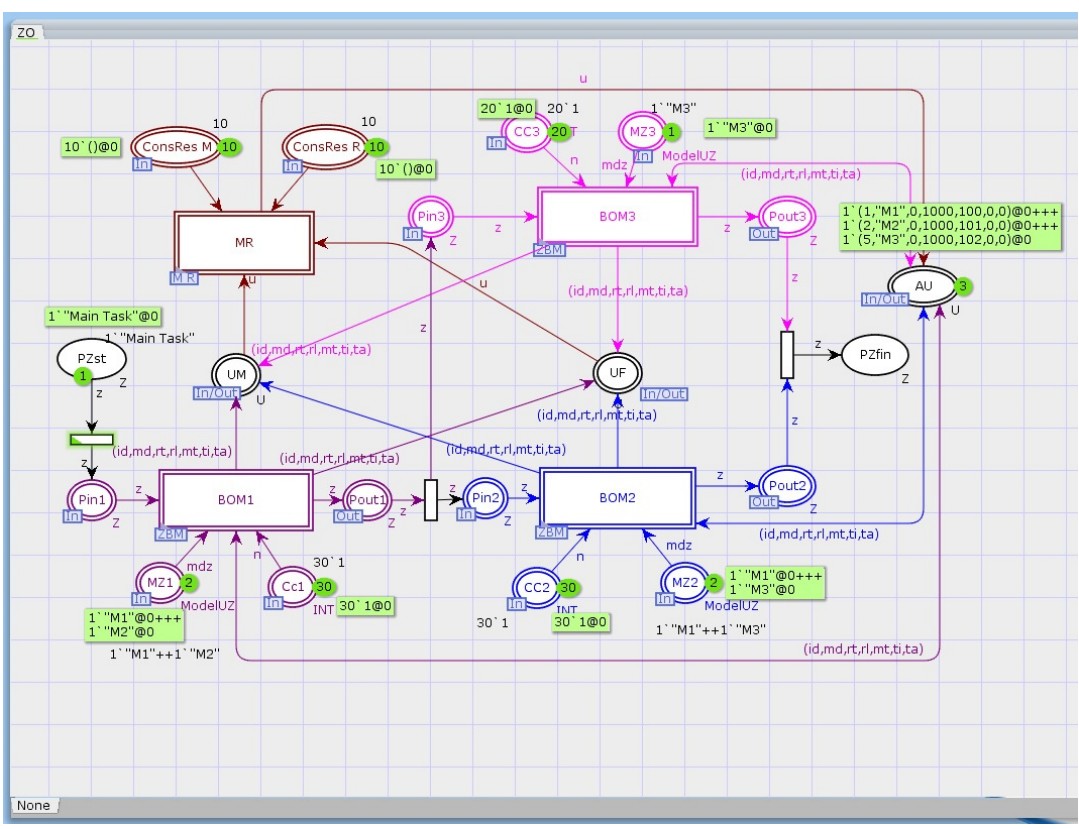

**Figure A3.** HPN AV initial marking.

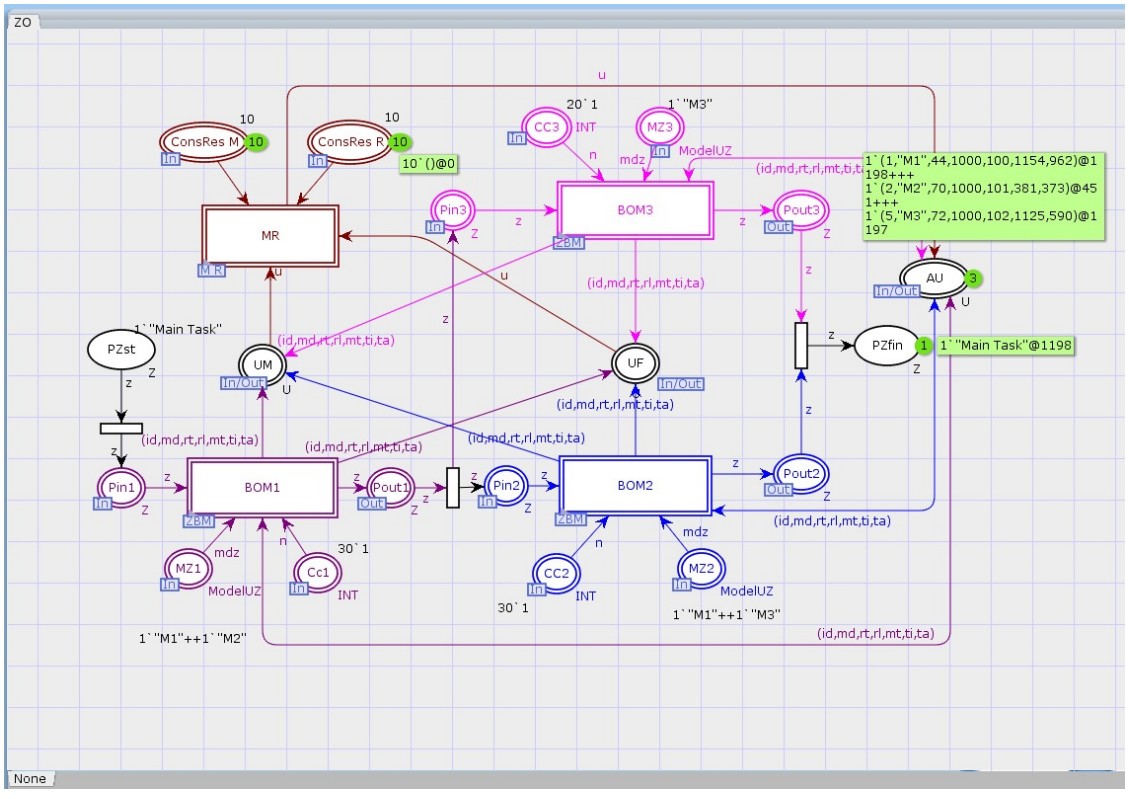

**Figure A4.** HPN AV final marking.

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
