# Peer review of "Application of Hierarchical Colored Petri Nets for Technological Facilities’ Maintenance Process Evaluation"

_applsci, doi:10.3390/app11115100_

Round 1

Reviewer 1 Report

In this paper, a hierarchical Petri Net based method was proposed to evaluate the maintenance and repairing efficiency on agricultural vehicles. The hierarchical simulation model was built using Timed Colored Petri Net; and was constructed with four levels of repair and maintenance modules. The practical utility of the model was then demonstrated via a robotic system of autonomous agricultural vehicles.

The proposed model is of scientific value to a wider area in the Industry 4.0. The demonstration of a real industrial case is a very good demonstration of the technical value of the proposed model. Technical details are sufficient in the paper which makes it easy to reproduce.

The writing should be improved in terms of its logical structure and wording, such that the scientific contribution of the work can be highlighted from an ocean of technical details. For example, the abstract should be rewritten to highlight the contribution of the work, and to summarise the work in a clear and concise manner. The conclusion should include, first, a conclusion of the proposed work, and then a discussion of possible future work. The authors should also indicate the generality of the proposed hierarchical Petri Net, and its limitations.

Author Response

For the point-by-point response to the reviewer’s comments please see the attachment.

Reviewer 2 Report

In references 26 and 27: replace "Scheng" by "Sheng"

Also please add to your references:

Le, B., & Andrews, J. (2016). Petri net modelling of bridge asset management using maintenance-related state conditions. Structure and Infrastructure Engineering12(6), 730-751.

Sheng, J., & Prescott, D. (2019). A coloured Petri net framework for modelling aircraft fleet maintenance. Reliability Engineering & System Safety189, 67-88.

Le, B., Andrews, J., & Fecarotti, C. (2017). A Petri net model for railway bridge maintenance. Proceedings of the Institution of Mechanical Engineers, Part O: Journal of Risk and Reliability231(3), 306-323.‏‏

Author Response

Response to Reviewer 2 Comments

Thank you for review of our paper applsci-1235523. We hope that the revised paper meets your expectations.

Here are answers to the reviewer's comments.

Point 1: In reference 26 and 27: replace "Scheng" by "Sheng"

Response 1: The name changed in text and references

Point 2: Also please add to your references:

Le, B., & Andrews, J. (2016). Petri net modelling of bridge asset management using maintenance-related state conditions. Structure and Infrastructure Engineering12(6), 730-751.

Sheng, J., & Prescott, D. (2019). A coloured Petri net framework for modelling aircraft fleet maintenance. Reliability Engineering & System Safety189, 67-88.

Le, B., Andrews, J., & Fecarotti, C. (2017). A Petri net model for railway bridge maintenance. Proceedings of the Institution of Mechanical Engineers, Part O: Journal of Risk and Reliability231(3), 306-323.

Response 2: Recommended articles added

Kind regards,

 Sergey Orlov

Samara State Technical University

Reviewer 3 Report

Dear Authors! Your article will undoubtedly arouse the interest of a certain circle  interested readers. However, I have a number of comments for you.
1. Any problem requires detailed and comprehensive study, namely, work with literary sources. Your bibliography is too small and does not fully reflect the full range of relevant research.
2. The conclusions are too general and I would like to see your contribution to the solution of the tasks, and not the prospects for further research.
3. The article is unnecessarily overloaded with tables and figures, it may make sense to place a part in the appendix.
4. Figure 9, I believe that there is no point in citing it, since there is nothing new in it.
5. It is necessary to observe the uniformity of terminology.

Author Response

For a point-by-point response to the reviewer’s comments please see the attachment.
